# Threshold Effects of New Energy Consumption Transformation on Economic Growth

**Fangming Xie, Chuanzhe Liu \*, Huiying Chen and Ning Wang**

School of Management, China University of Mining & Technology, Xuzhou 221116, China; m15246349728@163.com (F.X.); chy651201@163.com (H.C.); m15152462390@163.com (N.W.)
**\*** Correspondence: rdean@cumt.edu.cn; Tel.: +86-510-8359-0168

**Abstract:** This study uses data from seven countries with high energy consumption levels in 1997–2016 (i.e., the US, China, Japan, Canada, South Korea, Germany, and France) to establish a panel threshold model and analyze the multiple threshold effects of new energy consumption transformation on economic growth. Research results show the non-linear impact of new energy consumption transformation on economic growth. On the one hand, the transformation of new energy consumption will occasionally bring economic costs, thereby resulting in a negative impact on economic growth. On the other hand, economic cost occasionally disappears, thereby resulting in the positive impact of the transformation of new energy consumption on economic growth. This study proposes that economic cost is affected by the levels of research and development (R&D), economic development, and traditional energy dependence, therefore, we use these three variables as threshold variables. Threshold variable is essential in a panel threshold model. The behavioral varies of model can be predicted when threshold variable is at different ranges of levels. In other words, the behavior of panel threshold model may change as the level of threshold variable changes. In particular, when the R&D level is used as a threshold variable, the impact of new energy consumption transformation on economic growth will change from negative to positive as the level of R&D increases. We present a similar conclusion when the level of economic development is used as a threshold variable. However, when the level of traditional energy dependence is used as the threshold variable, the impact of new energy consumption transformation on economic growth will change from positive to negative as the level of traditional energy dependence increases.

**Keywords:** new energy consumption transformation; economic growth; economic cost; threshold effects

## 1. Introduction

New energy consumption is the amount of new energy or power used. The traditional energy consumption structure is mainly based on fossil energy. However, with the rapid development of the world economy, the intensity of energy consumption has gradually increased, and the increasing energy consumption has likewise caused bottlenecks in the supply of traditional fossil energy. To achieve sustainable development, countries have turned their attention to the development and utilization of new energy [1], while the global energy consumption structure has gradually transformed from traditional fossil energy to new energy sources. New energy is any energy source that is an alternative to fossil fuel. Therefore, new energy is considered an environment-friendly energy source and intends to address concerns about fossil fuels, such as its high carbon dioxide emissions, an important factor in global warming. This study uses the definition of new energy by the United Nations Development Programme (UNDP) as basis to classify water, nuclear, biomass, solar, wind, and other renewable energy as new energy category; Oil, coal, and natural gas as traditional fossil energy. In 2016, the global total new energy consumption was 2004 million tonne of oil equivalent (Million toe),

thereby accounting for approximately one-seventh of the world's total energy consumption. The 2018 BP International Energy Outlook [2] divides the energy transformation speed into three categories: evolving transformation (ET), faster transformation (FT), and even faster transition (EFT). This outlook also predicts the proportion of primary energy consumption in energy sources in the year 2040 under different energy transformation speeds. As shown in Figure 1, the proportion of new energy consumption will increase rapidly, new energy consumption will account for approximately one quarter, one-third, one-half respectively of the time of the total energy consumption in the year 2040 under three categories energy transformation speed.

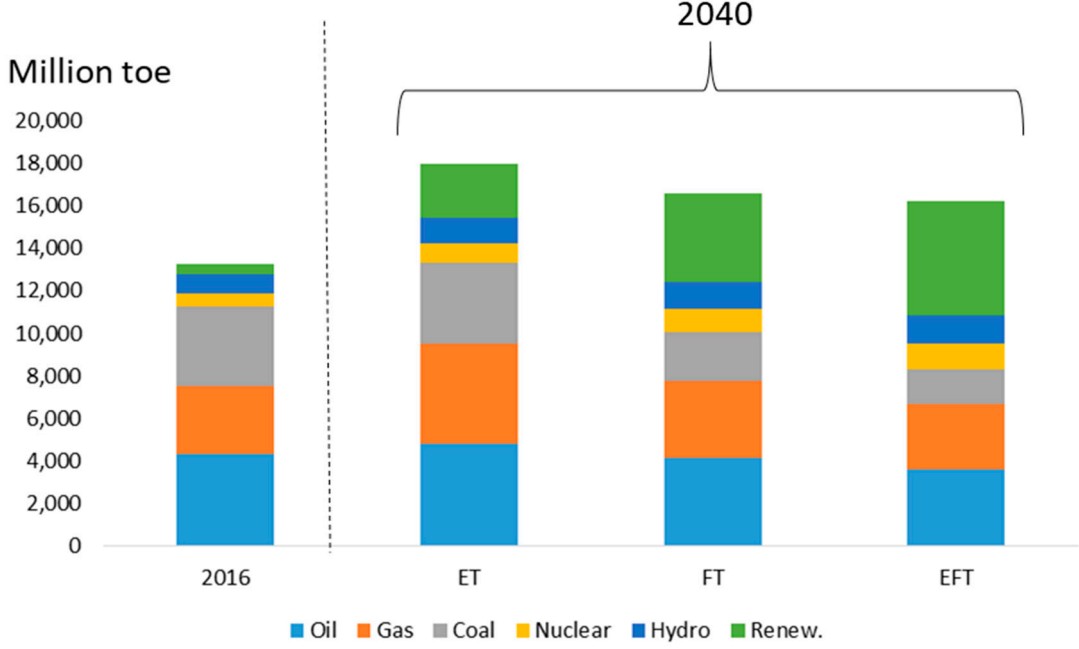

**Figure 1.** Forecast of the primary energy consumption by different energy types.

Energy is an indispensable element in boosting and sustaining the level of economic growth of a country. Supporting the direction of this assertion, Bernd and Wood (1975) [3], Rashe and Tatom (1977) [4] analyzed the relationship between energy consumption and economic growth earlier by using energy consumption as a production factor in the Cobb–Douglas production function, thereby proving a certain relationship between energy consumption and economic growth. Furthermore Ucan et al. (2014) [5], Rafindadi (2014) [6], and Rafindadi and Ozturk (2016) [7] argued that high energy consumption is one of the basic indicators of economic development level improved by a country.

As the current rapid transformation of new energy consumption, the energy consumption structure has also changed. Therefore, many studies have also turned their attention to the impact of new energy consumption on economic growth. However, the results of these studies are controversial and divergent. Accordingly, the following questions should be answered: Is the impact of new energy consumption transformation on economic growth positive or negative? Does this effect have differences between countries or regions? Answering these questions can provide scientific recommendations for better development of new energy, avoiding the adverse effects of new energy consumption transformation on the economy and achieving the goal of new energy consumption share in the Paris Agreement with minimal economic costs. The novelty of this paper is that it not only proves threshold effects of new energy consumption transformation on economic growth, but also finds the reasons for such threshold effects by using the panel threshold model.

The remainder of the paper is organized as follows. Section 2 introduces a literature review of studies about the impact of new energy consumption on economic growth. Section 3 combines three threshold variables (R&D level; level of economic development; and level of traditional energy

dependence) to analyze the threshold effects of new energy consumption on economic growth and proposed three hypotheses. Section 4 selects variables and builds panel threshold models. Section 5 analyzes the empirical results and confirms the three hypotheses presented in Section 3. Section 6 outlines conclusions and proposes recommendations.

## 2. Literature Review

At present, studies about impact of new energy consumption on economic growth mainly use linear and grouping methods from the perspective of different mechanisms. However, the results of these studies are controversial and divergent.

First, several studies have indicated that new energy consumption can promote economic growth. Fang (2011) [8], Tugcu et al. (2012) [9], Apergis and Payne (2012) [10], Shafiei and Salim (2014) [11], and Lotz (2016) [12] used new energy consumption as a factor of production and applied into Cobb–Douglas production function. Their conclusions indicated that new energy consumption can completely replace traditional energy consumption and promote economic growth like traditional energy consumption and other production factors. Therefore, promoting new energy consumption bears benefits not only for the environment but also for the economic conditions of the countries. Apergis and Payne (2010) [13], Lin and Moubarak (2014) [14], Rafindadi and Ozturk (2017) [15], and Marinas et al. (2018) [16] indicated that there is positive bi-directional long run relationship between new energy consumption and economic growth by using data from Organization for Economic Co-operation and Development (OECD) countries, China, Germany, and South Africa, respectively. The bi-directional relationship between new energy consumption and economic growth implied that growing economy is propitious for the development of new energy sector which in turn helps to boost economic growth. Tiwari (2011) [17] believed that the new energy consumption can promote economic growth in India. They also pointed out that the share of new energy consumption explained a significant part of the forecast error variance of GDP by using the structural vector autoregressive (SVAR) model. Bloch et al. (2015) [18] pointed out that Chinese growth is led by coal consumption, oil consumption, and new energy consumption. Economic growth also causes coal, oil, and new energy consumption. Nasen et al. (2016) [19] asserted that increasing the new energy consumption, the efficiency of the energies is increasing and leads to a high economic growth. Ito (2017) [20] and Fotourehchi (2017) [21] asserted that new energy consumption leads to a positive impact on economic growth for developing countries. Paramati et al. (2018) [22] pointed out that new and traditional energy consumptions have a significant positive effect on the economic activities across the sectors and the overall economic output. They also pointed out that the impact is more from new energy consumption on economic activities than that of traditional energy consumption. Saad and Taleb (2018) [23] pointed out that the impact of new energy consumption on economic growth is positive and the presence of uni-directional causality running from economic growth to new energy consumption in the short run, a bi-directional causal relationship between the variables in the long run. Ntanos et al. (2018) [24] revealed that there is a correlation between economic growth and new energy consumption and fossil energy consumption. Furthermore, they believed that there is a higher correlation between new energy consumption and the economic growth of countries with high Gross Domestic Product (GDP) levels than those with low GDP levels.

Second, some studies have indicated that the development of new energy technologies is continuously improving compared with traditional energy technologies. Therefore, the transformation of new energy consumption will result in relative economic costs, thereby leading to a negative impact on economic growth. Based on the autoregressive distributed lag model (ARDL), Ocal and Aslan (2013) [25] and Khoshnevis and Bahram (2017) [26] pointed out that new energy consumption has a negative impact on economic growth and there is the uni-directional causality from new energy consumption to economic growths. Sasana and Ghozali (2017) [27] got a similar conclusion by using data from Brazil, Russia, India, China, and South Africa. Magazzino (2017) [28] suggested that new energy consumption increases by 1%, real GDP decreases by 0.23% in Italy. They also pointed out that

the share of new energy consumption explained a significant part of the forecast error variance of real GDP by using Yamamoto approach.

Finally, with the expanding research, some studies have explained that regional heterogeneity may exist in the impact of new energy consumption on economic growth. Lee and Chang (2007) [29] and Qi and Li (2017) [30] used real GDP per capita as a basis for grouping different countries. The former asserted that bi-directional causality between the new energy consumption and economic growth in countries with high real GDP per capita, but there is uni-directional causality from new energy consumption to economic growth in countries with low real GDP per capita. The latter pointed out that new energy consumption has a positive impact on economic growth in countries with high real GDP per capita, but new energy consumption will result in economic costs and have a negative impact on economic growth in countries with low real GDP per capita. Their studies indicated that the regional heterogeneity of the impact of new energy consumption transformation on economic growth is caused by the level of real GDP per capita. Huang et al. (2008) [31] and AI-mulali and Fereidouni (2013) [32] divided countries into four categories (i.e., low-income, low-and-middle-income, middle-and-high-income, and high-income groups) to study the impact of new energy consumption on economic growth, but reached different conclusions. The former asserted that: there exists no causal relationship between new energy consumption and economic growth in the low income group; in the middle income groups, economic growth leads energy consumption positively; in the high income groups, economic growth leads energy consumption negatively. The latter pointed out that: in high-income group countries, there is a positive bi-directional long run relationship between new energy consumption and economic growth; in the middle income groups, there are no long run relationship between the variables; countries with low income level showed a uni-directional long run relationship from economic growth to new energy consumption. Their studies indicated that the regional heterogeneity of the impact of new energy consumption transformation on economic growth is caused by the level of income. Bhattacharya and Paramati (2015) [33] pointed out that in countries where significant shift towards new energy occurred during their study period, new energy consumption as a significant driver in economic growth (i.e., Canada, the Czech Republic, China, France, Germany, the United Kingdom, etc.). However, in other countries new energy consumption had a negative effect on economic growth (i.e., India, Ukraine and Israel). Alper and Oguz (2016) [34] used ARDL model to analyze the data from European Union, the results showed that: new energy consumption has positive impacts on economic growth for all countries, but there is statistically significant impact on economic growth only for Estonia, Bulgaria, Slovenia, and Poland. Destek (2016) [35] believed that new energy consumption has a negative impact on economic growth in South Africa and Mexico but has a positive impact on economic growth in India. Pata and Terzi (2017) [36] found a uni-directional causality moving from new energy consumption to economic growth in Germany and Japan, and a bi-directional causality between these two variables in France, Italy, and the United Kingdom by using the bootstrap panel Granger causality method.

The preceding studies mainly use linear or grouping methods to analyze the impact of new energy consumption transformation on economic growth. Several studies indicated that new energy consumption has a positive impact on economic growth, some studies asserted that new energy consumption is not conducive to economic growth, and other studies pointed that regional heterogeneity may exist in the impact of new energy consumption on economic growth. Since previous research did not reach a consistent conclusion, we suspect that the impact of new energy consumption transformation on economic growth may be non-linear, the results obtained using the linear method are inaccurate. While grouping methods can only obtain regional heterogeneity, although the reason for such heterogeneity cannot be determined. Therefore, this study innovates by using the panel threshold model to investigate the threshold effects of new energy consumption transitions on economic growth and determine the reasons for such threshold effects.

### 3. Theoretical Analysis of the Threshold Effect

We propose that the impact of new energy consumption transformation on economic growth may be non-linear. This study argues that the levels of R&D, economic development, and traditional energy dependence are the reasons for this non-linear existence. Therefore, this study uses the panel threshold model and levels of R&D, economic development, and traditional energy dependence as threshold variables to study the threshold effects of new energy consumption transformation on economic growth. The theoretical analyses of the different threshold variables are as follows.

*3.1. R&D Level*

The main reason for the economic cost of the new energy consumption transformation is the promotion of transformation when new energy technologies are insufficient. Some studies have indicated that new energy consumption can promote economic growth like traditional energy consumption and other production factors. This is because the substitution effect can be applied to energy consumption. Awerbuch and Sauter (2006) [37] pointed out that given the similarity between the production and use of new and traditional energy sources, the substitution effect of new energy consumption mainly refers to the impact of replacing traditional energy on economic growth. However, in some countries with low R&D levels, due to the backwardness of new energy technologies, the development of new energy consumption requires higher costs. Since the cost of using new energy is higher than the cost of using traditional energy, the substitution effect cannot make new energy consumption completely replace traditional energy consumption to promote economic growth. At this time, the transformation of new energy consumption will have a negative impact on economic growth. Compared with countries with low R&D levels, countries with high R&D levels have more sufficient research funds, scientific researchers, professional knowledge, and infrastructure and equipment to support the development of new energy technologies. The improvement of new energy technology level reduces the cost of using new energy, so the substitution effect can make new energy consumption completely replace traditional energy to promote economic growth. In general, countries with high R&D levels have good technologies to support the transformation of new energy consumption, thereby possibly reducing or even eliminating the economic cost of the new energy consumption transformation. By contrast, due to the backwardness of new energy technologies, countries with low R&D levels may face a huge economic cost of achieving new energy consumption transformation. Thus, this study proposes:

**Hypothesis 1:** *When the R&D level is below a certain threshold, the impact of new energy consumption transformation on economic growth is negative. When the R&D level is above a certain threshold, the impact may change from negative to positive.*

*3.2. Level of Economic Development*

Some studies have indicated that the regional heterogeneity of the impact of new energy consumption transformation on economic growth is caused by the level of real GDP per capita. Therefore we assume the level of economic development is one of the reasons for the economic costs of new energy consumption transformation. Batlle (2011) [38] found that new energy development costs are high and the government needs to support preferential policies, such as subsidies, taxes, and loans. When the government revenue is certain, the expenditure on new energy subsidies will have a crowding out effect on other expenditures, which may be detrimental to the country's economic growth. If the new energy development expenditure is excessive, the powerful crowding out effect may cause enormous economic costs. However people in countries with high levels of economic development are generally advanced in environmental protection and have high demand for new energy consumption. Thereby people in countries with high economic development levels can spontaneously promote the transformation of new energy consumption, which can reduce the government's expenditure for new energy development. In addition, countries with high levels of economic development are prone

to introducing the "combined effects", thereby easily attracting capital, technology, and high-tech talent for new energy development, which can also help reduce the economic cost of new energy consumption transformation. Hence, this study proposes:

**Hypothesis 2:** *When the level of economic development is below a certain threshold, the impact of new energy consumption transformation on economic growth is negative. When the level of economic development is above a certain threshold, the impact may change from negative to positive.*

### 3.3. Level of Traditional Energy Dependence

Traditional energy dependence is the application of path-dependence theory [39] to energy consumption. Path dependence means that once people choose a system, such factors as economies of scale and learning effect will cause the system to continuously strengthen itself along a certain direction. Once people make a choice, such decision is similar to the point of no return. Accordingly, the power of inertia will make this choice self-reinforcing and result in difficulty for individuals to back out. Economies of scale can increase the scale of production and reduce the long-term average cost in the production process. Meanwhile, the learning effect can reduce the marginal cost of human capital accumulation and technology diffusion in the production process. Zwaan and Rabl (2004) [40], Junginger et al. (2005) [41], and Kobos et al. (2006) [42] indicated that the increase in the scale of new energy can considerably form economies of scale and learning effects. However, development substantially exceeds the time of new energy development because of the time involved in traditional energy. New energy technologies remain immature. Energy consumption in several countries is considerably dependent on traditional energy sources, even energy consumption is locked on fossil energy. In these countries, producing economies of scale and learning effects is difficult in relation to new energy sources. Klitkou et al. (2015) [43] determined that mature energy technologies have a distinct advantage over new energy technologies, not because they are necessarily better, but because their positive feedback mechanisms create additional benefits and decrease costs of production. In other words, new energy technologies may increase the economic cost of production. Therefore, the development of new energy in areas with high dependence on traditional energy will face strong economic resistance, while the new energy consumption transformation will undoubtedly result in immense economic cost. Accordingly, this study proposes:

**Hypothesis 3:** *When the level of traditional energy dependence is above a certain threshold, the impact of new energy consumption transformation on economic growth is negative. When the level of traditional energy dependence is below a certain threshold, the impact may change from positive to negative.*

## 4. Selecting the Variables and Building the Models

This study focuses on the effects of new energy consumption transitions on economic growth and analyzes the non-linear effects of new energy consumption transformation on economic growth under different levels of R&D, economic development, and traditional energy dependence. According to the availability of data, this study selects seven countries with high energy consumption levels (i.e., the US, China, Japan, Canada, Korea, Germany, and France) for 10 years (i.e., 1997–2016). The definition of each variable is as follows:

Dependent Variable: Dependent variable is used in statistics as the outcome variable which one can be predicted by using independent variables. Because the economic growth is the outcome variable of our study, so we use economic growth as dependent variable. Economic growth (Y, unit: millions of US dollars) is expressed in terms of actual Gross Domestic Product (GDP) of each country;

Independent Variable: Independent variable is used in statistics to predict or explain dependent variable. We want to predict or explain economic growth through the level of new energy consumption transformation, therefore we use the level of new energy consumption transformation as independent

variable. The level of new energy consumption transformation (new, unit: percentage) is expressed as the ratio of new energy consumption (unit: Million toe) to total energy consumption (unit: Million toe).

Threshold Variable: Threshold variable is essential in a panel threshold model. We can study the behavior predicted by the model varies when threshold variable is at different ranges of levels. This paper uses the following three variables as threshold variable respectively to build the panel threshold model: Level of R&D (unit: percentage) is expressed as the ratio of national R&D expenditure (unit: millions of US dollars) to actual GDP (unit: millions of US dollars); level of economic development (ED, unit: US dollars) is expressed by the real GDP per capita of each country; level of traditional energy dependence (TED, unit: Million toe/millions of US dollars) is expressed as the ratio of the total amount of traditional energy consumption (unit: Million toe) to the actual GDP (unit: millions of US dollars) of each country;

Controlled Variable: Capital input (K, unit: percentage) expressed as the proportion of total capital formation (unit: millions of US dollars) in the actual GDP (unit: millions of US dollars) of a country; Labor input (L, unit: thousand people) expressed by the total number of employed of each country.

All data are logarithmically used to reduce the possible heteroscedasticity in the data except for those in units of percentage (Because the data are larger than 0 and less than 1 and do not have extreme outliers, there is no need to eliminate heteroscedasticity). The data used in this study are from the 2018 BP Statistical review of World Energy [43] and WIND database (A financial database provided by Wind information company). Table 1 shows the statistical description of each variable.

**Table 1.** Statistical description of variables.

| Variables/(Units) | Observation | Average Value | Standard Deviation | Minimum Value | Maximum Value |
|---|---|---|---|---|---|
| Y/(millions of US dollars) | 140 | 7.890 | 0.949 | 5.925 | 9.805 |
| New/(%) | 140 | 0.212 | 0.128 | 0.049 | 0.510 |
| R&D/(%) | 140 | 0.024 | 0.007 | 0.006 | 0.043 |
| ED/(US dollars) | 140 | 9.995 | 1.056 | 6.564 | 10.941 |
| TED/(Million toe/millions of US dollars) | 140 | 0.788 | 0.128 | 0.490 | 0.950 |
| K/(%) | 140 | 0.264 | 0.076 | 0.175 | 0.477 |
| L/(thousand people) | 140 | 11.048 | 1.217 | 9.630 | 13.576 |

This study uses the panel threshold model proposed by Hansen [44]. The panel threshold model can analyze the degree or direction of influence between variable changes after a certain threshold variable reaches a certain threshold value. The essence of this model is to incorporate the threshold variable as an unknown variable into the empirical model, construct a piecewise function, and estimate the threshold and parameters of the different threshold intervals. The basic equation of the model is as follows:

$$Y_{it} = \mu_i + \beta'_1 x_{it} I(q_{it} \leq \gamma) + \beta'_2 x_{it} I(q_{it} > \gamma) + e_{it} \tag{1}$$

where $i$ represents the country, $t$ represents the year, $Y_{it}$ represents the dependent variable, $x_{it}$ is the independent variable, $q_{it}$ represents the threshold variable, $\gamma$ is the threshold value to be estimated, $e_{it}$ is a random disturbance item, $\beta'_1$, $\beta'_2$ are the coefficient to be estimated for each variables, $\mu_i$ is to remove individual-specific means, and $I(.)$ is the indicator function, which is 1 when the conditions in the parentheses are true and 0 when the conditions in the parentheses are not established. Therefore, Formula (1) can also be expressed as follows:

$$Y_{it} = \begin{cases} \mu_i + \beta'_0 z_{it} + \beta'_1 x_{it} + e_{it}, q_{it} \leq \gamma \\ \mu_i + \beta'_0 z_{it} + \beta'_2 x_{it} + e_{it}, q_{it} \leq \gamma \end{cases}. \tag{2}$$

$z_{it}$ is a set of control variables. The panel threshold model is divided into two regions (regime), while the sum of the squared errors based on the Hansen calculations is as follows:

$$S_1(\gamma) = \hat{e}^*(\gamma)' \hat{e}^*(\gamma) = Y^{*\prime}(1 - x^*(\gamma)'(x^*(\gamma)'x^*(\gamma))^{-1}x^*(\gamma)')Y^* \tag{3}$$

where $x_{it}(\gamma) = \begin{pmatrix} x_{it}I(q \leq \gamma) \\ x_{it}I(q > \gamma) \end{pmatrix}$; $Y_{it}^* = Y_{it} - \overline{Y}_{it}$; $e_{it}^* = e_{it} - \overline{e}_{it}$; $x_{it}^* = x_{it} - \overline{x}_{it}$; $\hat{\beta}(\gamma) = (x^*(\gamma)'x^*(\gamma))^{-1}x^*(\gamma)'Y^*$; $\hat{e}^*(\gamma) = Y^* - x^*(\gamma)\hat{\beta}^*(\gamma)$. Moreover, $\gamma$ can be estimated by least squares and can be easily determined by minimization of the concentrated sum of the squared errors (3). The least squares estimators are as follows:

$$\hat{\gamma} = \underset{\gamma}{\operatorname{argmin}} S_1(\gamma). \tag{4}$$

After $\hat{\gamma}$ is obtained, the residual variance is as follows:

$$\hat{\sigma}^2 = \frac{1}{n(T-1)}\hat{e}^{*\prime}\hat{e}^* = \frac{1}{n(T-1)}S_1(\widehat{\gamma}). \tag{5}$$

This study will build the panel threshold model on the basis of the preceding variables. Thus, we need to rewrite Formula (1) in the following form.

The single threshold model (i.e., panel threshold model with only one threshold value) can be expressed as follows:

$$Y_{it} = \mu_i + \beta'_0 z_{it} + \beta'_1 new_{it}I(q_{it} \leq \gamma_1) + \beta'_2 new_{it}I(q_{it} > \gamma_1) + e_{it}. \tag{6}$$

Thereafter, we can deduce the multiple threshold model (i.e., panel threshold model with multiple threshold values) by using the double threshold model as an example and can be expressed as follows:

$$Y_{it} = \mu_i + \beta'_0 z_{it} + \beta'_1 new_{it}I(q_{it} \leq \gamma_1) + \beta'_2 new_{it}I(\gamma_1 < q_{it} \leq \gamma_2)$$
$$+ \beta'_3 new_{it}I(q_{it} > \gamma_2) + e_{it} \tag{7}$$

The panel threshold model can be built and run on Stata14.

## 5. Empirical Results and Analysis

In the context of equation (7) there are three cases: no thresholds (neither $\gamma_1$ nor $\gamma_2$ exist); the model has a threshold ($\gamma_1$ exists and $\gamma_2$ does not exist), the model has two thresholds ($\gamma_1$ and $\gamma_2$ are both exist); and model (7) is estimated on the basis of these three cases. The data are repeatedly sampled 300 times using the bootstrap method to obtain the definite F and P values (see Table 2). Table 2 shows that in the panel threshold model, the R&D has two threshold values and the first and second threshold values are significant at the 1% and 10% levels, respectively. ED also has two threshold values that are significant at the 1% level. TED has only one threshold value, which is significant at the 1% level.

**Table 2.** Threshold effect test.

| Variables | Single Threshold Model | | | | | | Double Threshold Model | | | | | |
|---|---|---|---|---|---|---|---|---|---|---|---|---|
| | F Value | *p* Value | BS | 10% | 5% | 1% | F Value | *p* Value | BS | 10% | 5% | 1% |
| R&D | 148.70 | 0.00 | 300 | 22.92 | 25.99 | 35.11 | 20.79 | 0.06 | 300 | 18.24 | 20.84 | 25.78 |
| ED | 162.49 | 0.00 | 300 | 35.85 | 41.77 | 66.21 | 50.67 | 0.00 | 300 | 25.17 | 31.50 | 47.04 |
| TED | 148.70 | 0.00 | 300 | 29.94 | 37.11 | 55.53 | 24.03 | 0.13 | 300 | 24.99 | 32.94 | 40.66 |

We can estimate the specific threshold values on the basis of the number of threshold value. Table 3 shows the estimated results. The model with two threshold values is divided into the following three segments: (1) when the level of threshold variable is lower than the first threshold value; (2) when the level of threshold variable is higher than the first threshold value but lower than the second threshold value; and (3) when the level of threshold variable is high than the second threshold value. The single threshold model is divided into two segments: (1) when the level of threshold variable is lower than the threshold value and (2) when the level of threshold variable is higher than the threshold value.

**Table 3.** Threshold value estimates.

| Variables | Threshold Value 1 [Lower Upper] | Threshold Value 2 [Lower Upper] |
|:---:|:---:|:---:|
| **R&D** | 0.009 [0.0071 0.0095] | 0.0140 [0.0135 0.0146] |
| **ED** | 7.4693 [7.2402 9.8142] | 8.9958 [8.9468 8.9978] |
| **TED** | 0.5695 [0.4362 0.6774] | |

The estimated results for each segment are as follows (see Table 4).

**Table 4.** Threshold model regression results.

| Y | Fixed Effect Model | Threshold Effect Model | | |
|:---:|:---:|:---:|:---:|:---:|
| | | **R&D** | **ED** | **TED** |
| **New** | 2.312 ***(0.004) | −25.010 *** (0.000) (A ≤ 0.009) | −11.116 *** (0.000) (ED ≤ 7.469) | 0.896 *** (0.006) (TR ≤ 0.5695) |
| | | −17.230 *** (0.000) (0.009 < A ≤ 0.014) | 1.694 *** (0.005) (7.469 < ED ≤ 8.996) | −18.647 *** (0.000) (TR > 0.5695) |
| | | 1.101 *** (0.031) (A > 0.014) | 5.439 *** (0.004) (ED > 8.996) | |
| **K** | 5.296 *** (0.000) | 0.191 (0.758) | 1.408 *** (0.048) | 1.011 (0.126) |
| **L** | 5.369 *** (0.000) | 5.067 *** (0.000) | 4.996 *** (0.000) | 5.348 *** (0.000) |
| **Cons** | −53.325 *** (0.000) | −48.279 *** (0.000) | −48.019 *** (0.000) | −51.958 *** (0.000) |
| **obs** | 140 | 140 | 140 | 140 |
| **F test** | 141.990 | 94.770 | 93.940 | 93.700 |
| **Prob > F** | 0.000 | 0.000 | 0.000 | 0.000 |
| **R²** | 0.712 | 0.889 | 0.843 | 0.863 |

Note: () is the standard error, "***" represent 1% levels of significance.

The results of the panel threshold model were established by using R&D, ED, and TED as threshold variables. Table 4 shows that the impact of new energy consumption transformation on economic growth are not the same in the different segments of the different threshold values. Therefore, the impact of new energy consumption transformation on economic growth have multiple threshold effects:

(1) Using the level of R&D as a threshold value. When the level of R&D is lower than the first threshold value (R&D = 0.009), the new energy consumption transformation has a significant negative impact on economic growth, while the marginal coefficient of the impact is −25.10. When the level of R&D is higher than the first threshold value but lower than the second threshold value (R&D = 0.014), the impact of the new energy consumption transformation on economic growth remains significantly negative. However, the negative impact is weakened, while the marginal coefficient of the impact is −17.203. When the level of R&D exceeds the second threshold value, the impact of the new energy consumption transformation on economic growth is significantly positive, while the marginal coefficient of the impact is 1.101. Overall, when the R&D level is used as a threshold variable, the impact of new energy consumption transformation on economic growth will change from negative to positive as the level of R&D increases. Therefore, Hypothesis 1 is confirmed. Evidently, the R&D level is an important factor that results in economic costs when the new energy consumption is transformed. Countries with low R&D level lack the sufficient technology to support the transformation of new energy consumption, thereby leading to a negative shift in economic growth owing to insufficient technology. By contrast, countries with high R&D level provide sufficient technical support for the transformation of new energy consumption, thereby new energy consumption can completely replace traditional energy consumption and promote economic growth along with traditional energy consumption.

(2) Using the level of economic development as a threshold variable. When the level of economic development is below the first threshold value (ED = 7.469), the new energy consumption

transformation has a significant negative impact on economic growth, while the marginal coefficient of the impact is $-11.116$. When the level of economic development is above the first threshold value but below the second threshold value (ED = 8.996), the impact of the new energy consumption transformation on economic growth is significantly positive, while the marginal coefficient of the impact is 1.694. When the level of economic development exceeds the second threshold value, the positive impact of the new energy consumption transformation on economic growth is amplified, while the marginal coefficient of the impact is 5.439. Overall, when the level of economic development is used as a threshold variable, the impact of new energy consumption transformation on economic growth will change from negative to positive as the level of economic development increases. Therefore, Hypothesis 2 is confirmed. Evidently, the level of economic development is also an important factor that results in the economic cost of the new energy consumption transformation. If the countries spends too much expenditure on new energy development, the powerful crowding out effect may cause enormous economic costs. However, people in countries with high levels of economic development are generally advanced in environmental protection and have high demand for new energy consumption. Thereby, people in countries with high economic development levels can spontaneously promote the transformation of new energy consumption, which can reduce the government's expenditure for new energy development. Moreover, countries with high level of economic developments are likely to attract capital, technology, and high-tech talents for new energy development. Thus, the economic cost of new energy consumption transformation can be reduced or even avoided in countries with high level of economic development.

(3) Using the level of traditional energy dependence as a threshold variable. When the level of traditional energy dependence is below the threshold value (TED = 0.5695), the new energy consumption transformation has a significant positive impact on economic growth, while the marginal coefficient of the impact is 0.896. However, when the level of traditional energy dependence is above the threshold value, the impact of new energy consumption transformation on economic growth is significantly negative, while the marginal coefficient of the impact is $-18.647$. In general, when the level of traditional energy dependence is used as the threshold variable, the impact of new energy consumption transformation on economic growth will change from positive to negative as the level of traditional energy dependence increases. Therefore, Hypothesis 3 is confirmed. Evidently, the level of traditional energy dependence is an important factor that results in the economic cost of the new energy consumption transformation. Hence, in countries with low levels of traditional energy dependence, new energy consumption can easily form economies of scale and learning effects and contribute to economic growth. By contrast, new energy consumption transformation in countries where energy consumption is mainly locked into fossil energy will inevitably generate considerable economic resistance and substantial economic costs, thereby leading to the negative impact of transformation on economic growth.

We also use the fixed-effects model to test the robustness of the panel threshold model (see Table 4). Apart from the coefficients of the new energy consumption transformation in the panel threshold model change between positive and negative as the threshold variable changes, the coefficients of all other variable are consistent with the fixed-effect and panel threshold models. This result indicates that the panel threshold model has good robustness.

## 6. Conclusions and Recommendations

This paper has researched the threshold effects of new energy consumption transformation on economic growth. Using panel data for seven countries with high levels of energy consumption (i.e., the US, China, Japan, Canada, South Korea, Germany, and France) over the period 1997–2016 to establish the panel threshold models. The results show that the threshold effects of the transformation of new energy consumption on economic growth is caused by the level of R&D, economic development, and traditional energy dependence. The overall conclusion is that: the impact of new energy consumption

transformation on economic growth is non-linear and it will change with the levels of R&D, economic development, and traditional energy dependence are at different ranges of levels.

In particular, (1) when the R&D level is below 0.009, the new energy consumption transformation will have a great negative impact on economic growth. When the level of R&D is higher than 0.009, but lower than 0.014, the negative impact of the new energy consumption transformation on economic growth will be weakened. When the level of R&D is higher than 0.014, the new energy consumption transformation will have a positive impact on economic growth. This conclusion indicates that the transformation of new energy consumption is not conducive to economic growth in countries with low R&D levels. On the contrary, the transformation of new energy consumption in countries with high R&D levels will promote economic growth.

(2) When the level of economic development is below than 7.469, the new energy consumption transformation will have a negative impact on economic growth. When the level of economic development is higher than 7.469 but lower than 8.996, the new energy consumption transformation will have a positive impact on economic growth. When the level of economic development is higher than 7.469, the positive impact is strengthened. This conclusion indicates that in countries with low levels of economic development, the transformation of new energy consumption will have a negative impact on economic growth. However, in countries with high levels of economic development, the transformation of new energy consumption can promote economic growth.

(3) When the level of traditional energy dependence is below than 0.5695, the new energy consumption transformation will have a positive impact on economic growth. When the level of traditional energy dependence is higher than 0.5695, the new energy consumption transformation will have a negative impact on economic growth. This conclusion indicates that the transformation of new energy consumption will have a negative impact on the economic growth of countries with high dependence on traditional energy sources. However, the transformation of new energy consumption in countries with low levels of traditional energy dependence can promote economic growth.

To achieve the goals of the Paris Agreement, the majority of the countries are actively guiding the transformation of new energy consumption and increasing emission reduction. However, the transformation of new energy consumption often results in certain economic costs. Occasionally, the transformation of new energy will even have a negative impact on economic growth, while blind transformation will increase the economic cost as well. Therefore, this study provides three recommendations by focusing on the research conclusions.

First, countries with high R&D levels can accelerate the transformation of new energy consumption. However, countries with low R&D levels should give priority to improving R&D levels before new energy consumption transformation. Because high R&D levels can ensure that the country has sufficient technical support for new energy consumption transformation. The main reason for the economic cost of the new energy consumption transformation is the promotion of transformation when new energy technologies are insufficient. Given the lack of sufficient research funds, scientific research personnel, professional knowledge stock, and infrastructure to support the transformation of new energy consumption, such transformation will bring immense economic costs and negatively impact the economic development in countries with low R&D levels. By contrast, countries with high R&D levels have sufficient new energy technologies. Accordingly, these countries can avoid economic costs and promote economic growth when pursuing the transformation of new energy consumption.

Second, countries with high levels of economic development can raise the level of new energy consumption, which is not only conducive to sustainable development, but also promotes national economic growth. Countries with low levels of economic development should be cautious in the transformation of new energy consumption, as this may be detrimental to the country's economic growth. The reason is that people in countries with high economic development levels can spontaneously promote the transformation of new energy consumption, which can reduce the government's expenditure for new energy development. In addition, countries with high levels

of economic development can easily attract capital, technology, and high-tech talent for new energy development. Therefore the new energy consumption transformation in countries with high levels of economic development can avoid economic costs. On the contrary, the transformation in countries with higher levels of economic development will have an adverse impact on economic growth.

Finally, for countries or regions with low levels of traditional energy dependence, the rate of new energy consumption transformation can be appropriately accelerated. For countries or regions where energy consumption has been "locked in" to fossil energy consumption, the transformation of new energy consumption should be approached with caution. Compared with traditional energy sources, the time of development for new energy sources is short, while the technologies of new energy sources have yet to mature. A few countries or regions have a high degree of dependence on traditional energy sources and their economic growth has a high dependence on traditional energy sources. The transformation of new energy consumption will face immense economic resistance and will have a substantial negative impact on economic growth. In countries or regions with low levels of dependence on traditional energy sources, new energy consumption can easily replace traditional energy consumption and have a positive impact on economic growth.

**Author Contributions:** Conceptualization, F.X.; Data curation, F.X.; Formal analysis, F.X.; Funding acquisition, C.L.; Investigation, F.X.; Methodology, F.X.; Resources, F.X. and C.L.; Software, F.X.; Supervision, C.L.; Validation, F.X.; Visualization, N.W.; Writing, original draft, F.X.; Writing, review, and editing, F.X., C.L., N.W., and H.C.

**Funding:** This research was funded by the "Double-First Class" Initiative Key Program of China University of Mining and Technology (Grant No. 2018WHCC07).

**Acknowledgments:** The authors are grateful for the financial support provided by the "Double-First Class" Initiative Key Program of China University of Mining and Technology (No.2018WHCC07).

**Conflicts of Interest:** The authors declare no conflict of interest. The funders had no role in the design of the study; collection, analyses, or interpretation of data; writing of the manuscript; and decision to publish the results.

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
