# Peer review of "Threshold Effects of New Energy Consumption Transformation on Economic Growth"

_sustainability, doi:10.3390/su10114124_

Round 1
Reviewer 1 Report
Thank you for your interesting work. There are a number of issues making necessary a revision. Introduction is not well written. It is hard to understand the main messages of each paragraph. It should be revised substantially. Literature review is very limited and does not comprehensively introduce and critically analyse the background of the subject. The paper should focus on related literature that has more closely connected with the present study though the authors refer to a lot of existing literature. Conclusions should be rewritten to understand the importance of research. The text should be edited following the journal template (see, for instance, references). Needs a reference to the equations.
Author Response
Response to Reviewers
Fangming Xie, Chuanzhe Liu, Huiying Chen and Ning Wang
At first, we appreciate editors’ and reviewers’ valuable comments and suggestions which help us improve the paper significantly. Following these comments and suggestions, we revised the paper as follows.
Response to Reviewer 1 Comments
Point 1: Introduction is not well written. It is hard to understand the main messages of each paragraph. It should be revised substantially.
Response 1: Thanks for the suggestion. According to this point, we have revised the introduction from four aspects.
First, we have revised the context of new energy consumption transformation in line 36-56, as that “New energy consumption is the amount of new energy or power used. The traditional energy consumption structure is mainly based on fossil energy. However, with the rapid development of the world economy, the intensity of energy consumption has gradually increased, and the increasing energy consumption has likewise caused bottlenecks in the supply of traditional fossil energy. To achieve sustainable development, countries have turned their attention to the development and utilization of new energy [1], while the global energy consumption structure has gradually transformed from traditional fossil energy to new energy sources. New energy is considered an environment-friendly energy source with new technologies and processes. This study uses the definition of new energy by the United Nations Development Programme (UNDP) as basis to classify water, nuclear, biomass, solar, wind, and other renewable energy as new energy category;Oil, coal, and natural gas as traditional fossil energy. In 2016, the global total new energy consumption was 2004 million tonne of oil equivalent (Million toe), thereby accounting for approximately one-seventh of the world's total energy consumption. The 2018 BP International Energy Outlook [2] divides the energy transformation speed into three categories: evolving transformation (ET), faster transformation (FT), and even faster transition (EFT). This outlook also predicts the proportion of primary energy consumption in energy sources in the year 2040 under different energy transformation speeds. The results show that the proportion of new energy consumption will increase rapidly, new energy consumption will account for approximately one quarter, one-third, one-half respectively of the time of the total energy consumption in the year 2040 under three categories energy transformation speed.”
Second, we briefly describe the relationship between energy consumption and economic growth in line 59-66, as that “Energy is an indispensable element in boosting and sustaining the level of economic growth of a country. Supporting the direction of this assertion,Bernd and Wood(1975) [3], Rashe and Tatom(1977) [4] analyzed the relationship between energy consumption and economic growth earlier by using energy consumption as a production factor in the Cobb–Douglas production function, thereby proving a certain relationship between energy consumption and economic growth. Furthermore Ucan et al. (2014) [5], Rafindadi (2014) [6], Rafindadi and Ozturk (2016) [7], argued that high energy consumption is one of the basic indicators of economic development level improved by a country.”
Third, Based on the above two points, we summarize the importance of studying new energy consumption for economic growth, mention the main purpose of the work and highlight the main novelty in line 67-77, as that “As the current rapid transformation of new energy consumption, the energy consumption structure has also changed. Therefore, many studies have also turned their attention to the impact of new energy consumption on economic growth. However the results of these studies are controversial and divergent. Accordingly, the following questions should be answered: Is the impact of new energy consumption transformation on economic growth positive or negative? Does this effect have differences between countries or regions? By answering these questions can help us achieve the Paris Agreement with minimal economic costs and avoid the adverse effects of blind new energy development on the economy. The novelty of this paper is that it not only proves threshold effects of new energy consumption transformation on economic growth, but also finds the reasons for such threshold effects by using the panel threshold model. ”
Finally, we have added an introduction to the main information of each paragraph in line 78-84, as that “The remainder of the paper is organized as follows. Section 2 introduces a literature review of studies about the impact of new energy consumption on economic growth. Section 3 combines three threshold variables (R&D level; level of economic development; level of traditional energy dependence) to analyze the threshold effects of new energy consumption on economic growth and proposed three hypotheses. Section 4 selects variables and builds panel threshold models. Section 5 analyzes the empirical results and confirms the three hypotheses presented in section 3. Section 6 outlines conclusions and proposes recommendations.”
Point 2: Literature review is very limited and does not comprehensively introduce and critically analyse the background of the subject. The paper should focus on related literature that has more closely connected with the present study though the authors refer to a lot of existing literature.
Response 2: Thanks for the suggestion. According to this point, we have carefully revised the literature review section. We have not only added related literatures, but also rearranged all literatures.
In line 86-88, we highlight controversial and diverging hypotheses, as that “At present, studies about the impact of new energy consumption on economic growth mainly use linear and grouping methods from the perspective of different mechanisms. However, the results of these studies are controversial and divergent.”
In line 89-112, we introduced some studies that support new energy consumption to promote economic growth, as that “First, several studies have indicated that new energy consumption can promote economic growth. Fang (2011) [8], Tugcu et al. (2012) [9], Apergis and Payne (2012) [10], Shafiei and Salim (2014) [11], Lotz (2016) [12] used new energy consumption as a factor of production and applied into Cobb-Douglas production function. Their conclusions indicated that new energy consumption can completely replace traditional energy consumption and promote economic growth like traditional energy consumption and other production factors. Apergis and Payne (2010)[13], Lin and Moubarak (2014)[14] , Rafindadi and Ozturk (2017)[15] ,Marinas et al. (2018) [16] indicated that there is positive bi-directional long run relationship between new energy consumption and economic growth by using data from Organization for Economic Co-operation and Development (OECD) countries, China, Germany and south Africa respectively. Tiwari (2011) [17] believed that the new energy consumption can promote economic growth in India by using the structural vector autoregressive (SVAR) model. Bloch et al. (2015) [18] pointed out that Chinese growth is led by coal consumption,oil consumption and new energy consumption. Nasen et al. (2016) [19] asserted that increasing the new energy consumption, the efficiency of the energies is increasing and leads to a high economic growth. Ito (2017) [20], Fotourehchi (2017) [21] asserted that new energy consumption leads to a positive impact on economic growth for developing countries. Paramati et al. (2018) [22] pointed out that new and traditional energy consumptions have a significant positive effect on the economic activities across the sectors and the overall economic output as well. Saad and Taleb (2018) [23] pointed out that the impact of new energy consumption on economic growth is positive. Moreover a uni-directional causality running from economic growth to new energy consumption in the short run , bi-directional causality between economic growth and new energy consumption in the short run. Ntanos et al. (2018) [24] revealed that there is a correlation between economic growth and new energy consumption and fossil energy consumption.”
In line 113-121, we introduced some studies that support new energy consumption has a negative impact on economic growth, as that “Second, some studies have indicated that the development of new energy technologies is continuously improving compared with traditional energy technologies. Therefore, the transformation of new energy consumption will result in relative economic costs, thereby leading to a negative impact on economic growth. Based on the autoregressive distributed lag model (ARDL), Ocal and Aslan (2013) [25], Khoshnevis and Bahram (2017) [26] pointed out that new energy consumption has a negative impact on economic growth in Turkey and Iran. Sasana and Ghozali (2017) [27] got a similar conclusion by using data from Brazil, Russia, India, China, and South Africa. Magazzino (2017) [28] used Yamamoto approach and suggested that new energy consumption increases by 1%, real GDP decreases by 0.23% in Italy.”
In line 122-155, we introduced some studies that support regional heterogeneity may exist in the impact of new energy consumption on economic growth, as that “Finally, with the expanding research, some studies have explained that regional heterogeneity may exist in the impact of new energy consumption on economic growth. Lee and Chang (2007) [29], Qi and Li (2017) [30] used real GDP per capita as a basis for grouping different countries. The former asserted that bi-directional causality between the new energy consumption and economic growth in countries with high real GDP per capita, but there is uni-directional causality from new energy consumption to economic growth in countries with low real GDP per capita. The latter pointed out that new energy consumption has a positive impact on economic growth in countries with high real GDP per capita, but new energy consumption will result in economic costs and have a negative impact on economic growth in countries with low real GDP per capita. Huang et al. (2008) [31], AI-mulali and Fereidouni (2013) [32] divided countries into four categories (i.e., low-income, low-and-middle-income, middle-and-high-income, and high-income groups) to study the impact of new energy consumption on economic growth, but reached different conclusions. The former asserted that: there exists no causal relationship between new energy consumption and economic growth in the low income group; in the middle income groups, economic growth leads energy consumption positively; in the high income groups, economic growth leads energy consumption negatively. The latter pointed out that: in high-income group countries, there is a positive bi-directional long run relationship between new energy consumption and economic growth; in the middle income groups, there are no long run relationship between the variables; countries with low income level showed a uni-directional long run relationship from economic growth to new energy consumption. Bhattacharya and Paramati (2015) [33] pointed out that in countries where significant shift towards new energy occurred during their study period, new energy consumption as a significant driver in economic growth (i.e., Canada, the Czech Republic, China, France, Germany, the United Kingdom, etc.). However in other countries, new energy consumption had a negative effect on economic growth (i.e., India, Ukraine, Israel and Israel). Alper and Oguz (2016) [34] used ARDL model to analyze the data from European Union, the results showed that: new energy consumption has positive impacts on economic growth for all countries, but there is statistically significant impact on economic growth only for Estonia, Bulgaria, Slovenia and Poland. Destek (2016) [35] believed that new energy consumption has a negative impact on economic growth in South Africa and Mexico but has a positive impact on economic growth in India. Pata and Terzi (2017) [36] used data from G7 countries found uni-directional causality moving from new energy consumption to economic growth in Germany and Japan, and a bi-directional causality between these two variables in France, Italy and the United Kingdom.”
At the end of this section, we provide a comprehensive introduction and critical analysis of the background of the topic, in addition to introducing the innovation of this paper in line 156-167, as that “The preceding studies mainly use linear or grouping methods to analyze the impact of new energy consumption transformation on economic growth. Some studies believed that new energy consumption has a positive impact on economic growth, some studies asserted that new energy consumption is not conducive to economic growth, and other studies pointed that regional heterogeneity may exist in the impact of new energy consumption on economic growth. Since previous research did not reach a consistent conclusion, we suspect that the impact of new energy consumption transformation on economic growth may be non-linear, the results obtained using the linear method are inaccurate. While grouping methods can only obtain regional heterogeneity, although the reason for such heterogeneity cannot be determined. Therefore, this study innovates by using the panel threshold model to investigate the threshold effects of new energy consumption transitions on economic growth and determine the reasons for such threshold effects.”
Point 3: Conclusions should be rewritten to understand the importance of research.
Response 3: Thanks for the suggestion. We rewrite the conclusion and present the original contribution of the research by focusing on the research results.
In line 418-426, we have summarized an overall conclusion as that “This paper has researched the threshold effects of new energy consumption transformation on economic growth .Using panel data for seven countries with high levels of energy consumption (i.e., the US, China, Japan, Canada, South Korea, Germany, and France) over the period 1997-2016 to establish the panel threshold models. The results show that the threshold effects of the transformation of new energy consumption on economic growth is caused by the level of R&D, economic development and traditional energy dependence .The overall conclusion is that: the impact of new energy consumption transformation on economic growth is non-linear and it will change with the levels of R&D, economic development and traditional energy dependence are at different ranges of levels.”
In line 427-435, we have summarized a conclusion when the R&D level is used as a threshold variable, as that “when the R&D level is below than 0.009, the new energy consumption transformation will have a great negative impact on economic growth. When the level of R&D is higher than 0.009 but lower than 0.014, the negative impact of the new energy consumption transformation on economic growth will be weakened. When the level of R&D is higher than 0.014, the new energy consumption transformation will have a positive impact on economic growth. This conclusion indicates that the transformation of new energy consumption is not conducive to economic growth in countries with low R&D levels. On the contrary, the transformation of new energy consumption in countries with high R&D levels will promote economic growth.”
In line 436-444, we have summarized a conclusion when the level of economic development is used as a threshold variable, as that “When the level of economic development is below than 7.469, the new energy consumption transformation will have a negative impact on economic growth. When the level of economic development is higher than 7.469 but lower than 8.996, the new energy consumption transformation will have a positive impact on economic growth. When the level of economic development is higher than 7.469, the positive impact is strengthened. This conclusion indicates that in countries with low levels of economic development, the transformation of new energy consumption will have a negative impact on economic growth. However, in countries with high levels of economic development, the transformation of new energy consumption can promote economic growth.”
In line 445-452, we have summarized a conclusion when the level of traditional energy dependence is used as a threshold variable, as that “When the level of traditional energy dependence is below than 0.5695, the new energy consumption transformation will have a positive impact on economic growth. When the level of traditional energy dependence is higher than 0.5695, the new energy consumption transformation will have a negative impact on economic growth. This conclusion indicates that the transformation of new energy consumption will have a negative impact on the economic growth of countries with high dependence on traditional energy sources. However, the transformation of new energy consumption in countries with low levels of traditional energy dependence can promote economic growth.”
In order to couple the conclusions and recommendations more suitably, we have made revision in line 459-462, 471-475 and 483-486 as that
“First, countries with high R&D levels can accelerate the transformation of new energy consumption. However, countries with low R&D levels should give priority to improving R&D levels before new energy consumption transformation. Because high R&D levels can ensure that the country has sufficient technical support for new energy consumption transformation.”
“Second, countries with high levels of economic development can raise the level of new energy consumption, which is not only conducive to sustainable development, but also promotes national economic growth. Countries with low levels of economic development should be cautious in the transformation of new energy consumption, as this may be detrimental to the country’s economic growth.”
“Finally, for countries or regions with low levels of traditional energy dependence, the rate of new energy consumption transformation can be appropriately accelerated. For countries or regions where energy consumption has been “locked in” to fossil energy consumption, the transformation of new energy consumption should be approached with caution.”
Point 4: The text should be edited following the journal template (see, for instance, references). Needs a reference to the equations.
Response 4: Thanks for the suggestion. We apology for the mistake. We re-edited the entire paper by following the journal template, including formats, figures, tables, equations and references.

Reviewer 2 Report
The manuscript describes the impact of the new energy consumption transitions on economic growth in seven countries with high levels of energy consumption. The authors claim that the economic cost is affected by the levels of R&D, the economic development, and the traditional energy dependence. From the overall presentation I would say that an interesting research work has been done.
However, some concerns arise regarding the manuscript. The research questions as well as the original contribution of the work, comparing to other previous works are not adequately presented. The authors should place more emphasis on (a) the research questions, (b) the coupling between the theoretical and the experimental analysis and (c) the original approach to the analysis of the problem.
Additional comments and recommendations for the improvement of the manuscript:
General notes
Ø All the acronyms should be explained within the manuscript.
Ø The English language should be improved throughout the manuscript.
Title
Ø The term “new energy consumption” should be explained within the text.
Abstract
Ø “In particular, if the R&D level is used as a threshold value but such level is below a particular threshold, then...”. This sentence should be improved.
Ø “threshold variable”. Please explain this term.
Ø “...then the positive effects can become negative”. What do the authors mean?
1. Introduction
Ø “… and utilization of new energy [1]” instead of “… and utilization of new energy [1]”.
Ø “In 2016, the global total energy consumption was 2004 million tonne of oil equivalent (Mtoe), thereby accounting for approximately one-seventh of the world's total energy consumption”. This sentence should be further explained.
Ø “The 2018 BP World Energy Outlook divides…”. A reference is missing here.
Ø “Figure 1 source: BP World Energy Outlook, 2018”. In-text citations usually include the reference number enclosed in square brackets or the author names in running text and the date of publication inside parentheses.
Ø Fig. 1. Units are missing.
Ø “Bernd and Wood (1975) and Rashe and Tatom (1977)”. References should be written in the same way throughout the manuscript.
2. Literature review
Ø General note: A more critical literature review is required.
Ø “…the existing studies has yet to reach…”. References are needed here.
Ø “Paramati, Apergis, and Ummalla (2018)”. References should be written in the same way throughout the manuscript.
Ø “Marinas, Dinu, and Socol (2018)”. Same comment as above.
Ø “Therefore, the current study innovates by using the panel threshold model to analyze the impact of new energy consumption transformation on economic growth”. This sentence is not clear and it should be further explained.
3. Theoretical analysis of the threshold effect
Ø “The preceding literature review indicates that the impact of new energy consumption transformation on economic growth may be non-linear”. Is that shown in the literature review?
Ø “Awerbuch and Sauter, 2006) [21]”. In-text citations usually include the reference number enclosed in square brackets or the author names in running text and the date of publication inside parentheses.
Ø “…countries with low R&D levels are economically costly to realize the transformation of new energy consumption owing to the backward new energy technologies”. This sentence is not clear.
Ø “…costs are mainly borne…”. This phrase is not clear.
Ø “…to new energy sources…” instead of “…to ne energy sources…”.
Ø “…Unruh (2000) determined that the industrial economy has been locked in by traditional fossil fuel-based energy sources [27]”. This reference is not recent.
4. Selecting the variables and building the models
Ø General note: All the equations parameters should be explained within the text.
Ø “Dependent variable” and “explanatory variable”. These terms are not adequately explained.
Ø “All data (except for those in units of percentage) are logarithmically used to reduce the possible heteroscedasticity in the data”. This sentence should be further explained.
Ø Table 1. Units are missing.
Ø Eq. 1. All the equation symbols should be written in the same way. E.g. “xit” and “xit”.
Ø “…R&D (R&D)…”. This is not correct.
5. Empirical results and analysis
Ø Table 2 should not be placed at the beginning of section 5.
Ø “We assume that the model has no thresholds, the model has a threshold, the model has two thresholds, and model (2) is estimated on the basis of the preceding three cases”. This sentence is not clear.
Ø “(1) when the threshold variable lower than the low threshold value…”. This sentence is not clear.
6. Conclusions and recommendations
Ø “This study first proposed three hypotheses”. What do the authors mean?
Ø In this section, the original contribution of the research has to be presented by focusing on the research results based on the research questions.
Author Response
Response to Reviewers
Fangming Xie, Chuanzhe Liu, Huiying Chen and Ning Wang
At first, we appreciate editors’ and reviewers’ valuable comments and suggestions which help us improve the paper significantly. Following these comments and suggestions, we revised the paper as follows.
Response to Reviewer 2 Comments
Point 1: The authors should place more emphasis on the research questions
Response 1: Thanks for the suggestion. In order to emphasize the research questions, we have made substantial revise to the introduction.
First, we have revised the context of the research questions, as that “New energy consumption is the amount of new energy or power used. The traditional energy consumption structure is mainly based on fossil energy. However, with the rapid development of the world economy, the intensity of energy consumption has gradually increased, and the increasing energy consumption has likewise caused bottlenecks in the supply of traditional fossil energy. To achieve sustainable development, countries have turned their attention to the development and utilization of new energy [1], while the global energy consumption structure has gradually transformed from traditional fossil energy to new energy sources. New energy is considered an environment-friendly energy source with new technologies and processes. This study uses the definition of new energy by the United Nations Development Programme (UNDP) as basis to classify water, nuclear, biomass, solar, wind, and other renewable energy as new energy category;Oil, coal, and natural gas as traditional fossil energy. In 2016, the global total new energy consumption was 2004 million tonne of oil equivalent (Million toe), thereby accounting for approximately one-seventh of the world's total energy consumption. The 2018 BP International Energy Outlook [2] divides the energy transformation speed into three categories: evolving transformation (ET), faster transformation (FT), and even faster transition (EFT). This outlook also predicts the proportion of primary energy consumption in energy sources in the year 2040 under different energy transformation speeds. The results show that the proportion of new energy consumption will increase rapidly, new energy consumption will account for approximately one quarter, one-third, one-half respectively of the time of the total energy consumption in the year 2040 under three categories energy transformation speed.”
Second, we briefly describe the relationship between energy consumption and economic growth in line 59-66, as that “Energy is an indispensable element in boosting and sustaining the level of economic growth of a country. Supporting the direction of this assertion,Bernd and Wood(1975) [3], Rashe and Tatom(1977) [4] analyzed the relationship between energy consumption and economic growth earlier by using energy consumption as a production factor in the Cobb–Douglas production function, thereby proving a certain relationship between energy consumption and economic growth. Furthermore Ucan et al. (2014) [5], Rafindadi (2014) [6], Rafindadi and Ozturk (2016) [7], argued that high energy consumption is one of the basic indicators of economic development level improved by a country.”
Third, Based on the above two points, we summarize the importance of research question and highlight the main novelty of this paper in line 67-77, as that “As the current rapid transformation of new energy consumption, the energy consumption structure has also changed. Therefore, many studies have also turned their attention to the impact of new energy consumption on economic growth. However the results of these studies are controversial and divergent. Accordingly, the following questions should be answered: Is the impact of new energy consumption transformation on economic growth positive or negative? Does this effect have differences between countries or regions? By answering these questions can help us achieve the Paris Agreement with minimal economic costs and avoid the adverse effects of blind new energy development on the economy. The novelty of this paper is that it not only proves threshold effects of new energy consumption transformation on economic growth, but also finds the reasons for such threshold effects by using the panel threshold model. ”
Point 2: The authors should place more emphasis on the coupling between the theoretical and the experimental analysis
Response 2: Thanks for the suggestion. According to this point, we reviews the description of theoretical and experimental analysis.
Line 178-202, as that “Some studies have indicated that new energy consumption can promote economic growth like traditional energy consumption and other production factors. This is because the substitution effect can be applied to energy consumption. Awerbuch and Sauter (2006) [37] pointed out that given the similarity between the production and use of new and traditional energy sources, the substitution effect of new energy consumption mainly refers to the impact of replacing traditional energy on economic growth. However, in some countries with low R&D levels, due to the backwardness of new energy technologies, the development of new energy consumption requires higher costs. Since the cost of using new energy is higher than the cost of using traditional energy, the substitution effect cannot make new energy consumption completely replace traditional energy consumption to promote economic growth. At this time, the transformation of new energy consumption will have a negative impact on economic growth. Compared with countries with low R&D levels, countries with high R&D levels have more sufficient research funds, scientific researchers, professional knowledge, and infrastructure and equipment to support the development of new energy technologies. The improvement of new energy technology level reduces the cost of using new energy, so the substitution effect can make new energy consumption completely replace traditional energy to promote economic growth. In general, countries with high R&D levels have good technologies to support the transformation of new energy consumption, thereby possibly reducing or even eliminating the economic cost of the new energy consumption transformation. By contrast, due to the backwardness of new energy technologies, countries with low R&D levels may face a huge economic cost of achieving new energy consumption transformation. Thus, this study proposes hypothesis 1: When the R&D level is below a certain threshold, the impact of new energy consumption transformation on economic growth is negative. When the R&D level is above a certain threshold, the impact may change from negative to positive.” can couple to line 361-363, as that “And “Overall, when the R&D level is used as a threshold variable, the impact of new energy consumption transformation on economic growth will change from negative to positive as the level of R&D increases. Therefore, Hypothesis 1 is confirmed.””
Line 204-223, as that “Some studies have indicated that the regional heterogeneity of the impact of new energy consumption transformation on economic growth is caused by the level of real GDP per capita. Therefore we assume the level of economic development is one of the reasons for the economic costs of new energy consumption transformation. Batlle (2011) [38] found that new energy development costs are high and the government needs to support preferential policies, such as subsidies, taxes, and loans. When the government revenue is certain, the expenditure on new energy subsidies will have a crowding out effect on other expenditures, which may be detrimental to the country's economic growth. If the new energy development expenditure is excessive, the powerful crowding out effect may cause enormous economic costs. However people in countries with high levels of economic development are generally advanced in environmental protection and have high demand for new energy consumption. Thereby people in countries with high economic development levels can spontaneously promote the transformation of new energy consumption, which can reduce the government's expenditure for new energy development. In addition, countries with high levels of economic development are prone to introducing the “combined effects”, thereby easily attracting capital, technology, and high-tech talent for new energy development, which can also help reduce the economic cost of new energy consumption transformation. Hence, this study proposes hypothesis 2: When the level of economic development is below a certain threshold, the impact of new energy consumption transformation on economic growth is negative. When the level of economic development is above a certain threshold, the impact may change from negative to positive.” can couple to line 380-383, as that “Overall, when the level of economic development is used as a threshold variable, the impact of new energy consumption transformation on economic growth will change from negative to positive as the level of economic development increases. Therefore, Hypothesis 2 is confirmed.”
Line 240-250, as that “Klitkou et al. (2015) [43] determined that mature energy technologies have a distinct advantage over new energy technologies, not because they are necessarily better, but because their positive feedback mechanisms create additional benefits and decrease costs of production. In other words, new energy technologies may increase the economic cost of production. Therefore, the development of new energy in areas with high dependence on traditional energy will face strong economic resistance, while the new energy consumption transformation will undoubtedly result in immense economic cost. Accordingly, this study proposes hypothesis 3: When the level of traditional energy dependence is above a certain threshold, the impact of new energy consumption transformation on economic growth is negative. When the level of traditional energy dependence is below a certain threshold, the impact may change from positive to negative.” can couple to line 401-404, as that “In general, when the level of traditional energy dependence is used as the threshold variable, the impact of new energy consumption transformation on economic growth will change from positive to negative as the level of traditional energy dependence increases. Therefore, Hypothesis 3 is confirmed.”
Point 3: The authors should place more emphasis on the original approach to the analysis of the problem.
Response 3: Thanks for the suggestion. According to this point, we provide a critical analysis of previous literature and emphasize the significance of using the panel threshold model to analyze this problem in line 156-167, as that “The preceding studies mainly use linear or grouping methods to analyze the impact of new energy consumption transformation on economic growth. Some studies believed that new energy consumption has a positive impact on economic growth, some studies asserted that new energy consumption is not conducive to economic growth, and other studies pointed that regional heterogeneity may exist in the impact of new energy consumption on economic growth. Since previous research did not reach a consistent conclusion, we suspect that the impact of new energy consumption transformation on economic growth may be non-linear, the results obtained using the linear method are inaccurate. While grouping methods can only obtain regional heterogeneity, although the reason for such heterogeneity cannot be determined. Therefore, this study innovates by using the panel threshold model to investigate the threshold effects of new energy consumption transitions on economic growth and determine the reasons for such threshold effects.”
Point 4: All the acronyms should be explained within the manuscript.
Response 4: Thanks for the suggestion. We have checked all the acronyms in this paper to ensure there is no undefined acronym in the revised manuscript.
Point 5: The English language should be improved throughout the manuscript.
Response 5: Thanks for the suggestion, our English language do need to be improved. Through re-reading of the full paper carefully, we have found a lot of phrases and grammatical errors, all of the errors have been revised. I hope it is clearer and accurate now of this revised paper on the English expression.
Point 6: The term “new energy consumption” should be explained within the text.
Response 6: Thanks for the suggestion. We have made revision in line 36, as that “New energy consumption is the amount of new energy or power used.”
Point 7:“In particular, if the R&D level is used as a threshold value but such level is below a particular threshold, then...” This sentence should be improved.
Response 7: Thanks for the suggestion. We have made revision in line 25-28, as that “In particular, when the R&D level is used as a threshold variable, the impact of new energy consumption transformation on economic growth will change from negative to positive as the level of R&D increases.”
Point 8: “threshold variable”. Please explain this term.
Response 8: Thanks for the suggestion. We have made revision in line 23-25, as that “Threshold variable is essential in a panel threshold model, it can help us to study the behavior predicted by the model varies when threshold variable is at different ranges of levels.”
Point 9: “...then the positive effects can become negative”. What do the authors mean?
Response 9: Thanks for the suggestion. We have made revision in line 29-32, as that “However, when the level of traditional energy dependence is used as the threshold variable, the impact of new energy consumption transformation on economic growth will change from positive to negative as the level of traditional energy dependence increases.”
Point 10: “… and utilization of new energy [1]” instead of “… and utilization of new energy [1]”.
Response 10: Thanks for the suggestion. We have made revision in line 40-41, as that “To achieve sustainable development, countries have turned their attention to the development and utilization of new energy [1],”.Moreover, we re-edited the entire paper by following the journal template, including formats, figures, tables, equations and references.
Point 11: “In 2016, the global total energy consumption was 2004 million tonne of oil equivalent (Mtoe), thereby accounting for approximately one-seventh of the world's total energy consumption”. This sentence should be further explained.
Response 11: Thanks for the suggestion. We have made revision in line 47-49, as that “In 2016, the global total new energy consumption was 2004 million tonne of oil equivalent (Million toe), thereby accounting for approximately one-seventh of the world's total energy consumption.”
Point 12: “The 2018 BP World Energy Outlook divides…” A reference is missing here.
Response 12: Thanks for the suggestion. We have made revision in line 507, as that “2. International Energy Outlook, 〈www.eia.gov/ieo/pdf/0484〉; 2018..”
Point 13: “Figure 1 source: BP World Energy Outlook, 2018”. In-text citations usually include the reference number enclosed in square brackets or the author names in running text and the date of publication inside parentheses.
Response 13: Thanks for the suggestion. We removed this sentence because the data is from BP International Energy Outlook and this outlook is listed in the reference.
Point 14: Fig. 1. Units are missing.
Response 14: Thanks for the suggestion. We have redesigned Figure 1, and the unit of Figure 1 is “million toe”, please see page 2.
Point 15: “Bernd and Wood (1975) and Rashe and Tatom (1977)”. References should be written in the same way throughout the manuscript.
Response 15: Thanks for the suggestion. We have made revision in line 60-61, as that “Bernd and Wood (1975) [3], Rashe and Tatom(1977) [4]” We review all references by this format.
Point 16: A more critical literature review is required.
Response 16: Thanks for the suggestion. According to this point, we have carefully revised the literature review section. We have not only added literatures(please see section 2), but also provide a comprehensive introduction and critical analysis of the background of the topic, in addition to introducing the innovation of this paper in line 156-167, as that “The preceding studies mainly use linear or grouping methods to analyze the impact of new energy consumption transformation on economic growth. Some studies believed that new energy consumption has a positive impact on economic growth, some studies asserted that new energy consumption is not conducive to economic growth, and other studies pointed that regional heterogeneity may exist in the impact of new energy consumption on economic growth. Since previous research did not reach a consistent conclusion, we suspect that the impact of new energy consumption transformation on economic growth may be non-linear, the results obtained using the linear method are inaccurate. While grouping methods can only obtain regional heterogeneity, although the reason for such heterogeneity cannot be determined. Therefore, this study innovates by using the panel threshold model to investigate the threshold effects of new energy consumption transitions on economic growth and determine the reasons for such threshold effects.”
Point 17: “…the existing studies has yet to reach…” References are needed here.
Response 17: Thanks for the suggestion. This is the mistake of our expression. We have made revision in line 86-88, we use “At present, studies about impact of new energy consumption on economic growth mainly use linear and grouping methods from the perspective of different mechanisms. However, the results of these studies are controversial and divergent.” to instead of the original expression.
Point 18: “Paramati, Apergis, and Ummalla (2018)”. References should be written in the same way throughout the manuscript.
Response 18: Thanks for the suggestion. We have made revision in line 105, as that “Paramati et al. (2018).” We review all references by this format.
Point 19: “Marinas, Dinu, and Socol (2018)”. Same comment as above.
Response 19: Thanks for the suggestion. We have made revision in line 95, as that “Marinas et al. (2018)”
Point 20: “Therefore, the current study innovates by using the panel threshold model to analyze the impact of new energy consumption transformation on economic growth”. This sentence is not clear and it should be further explained.
Response 20: Thanks for the suggestion. We have made revision in line 165-167, as that “Since previous research did not reach a consistent conclusion, we suspect that the impact of new energy consumption transformation on economic growth may be non-linear, the results obtained using the linear method are inaccurate. While the group methods can only obtain regional heterogeneity, although the reason for such heterogeneity cannot be determined. Therefore, this study innovates by using the panel threshold model to analyze threshold effects of new energy consumption transitions on economic growth and determine the reasons for such threshold effects.”
Point 21: “The preceding literature review indicates that the impact of new energy consumption transformation on economic growth may be non-linear”. Is that shown in the literature review?
Response 21: Thanks for the suggestion. This is the mistake of our expression. We have made revision in line 161-167, we use “We propose that the impact of new energy consumption transformation on economic growth may be non-linear.” to instead of the original expression.
Point 22: “Awerbuch and Sauter, 2006) [21]”. In-text citations usually include the reference number enclosed in square brackets or the author names in running text and the date of publication inside parentheses.
Response 22: Thanks for the suggestion. We have made revision in line 181, as that “Awerbuch and Sauter (2006) [37]”
Point 23: “…countries with low R&D levels are economically costly to realize the transformation of new energy consumption owing to the backward new energy technologies”. This sentence is not clear.
Response 23: Thanks for the suggestion. We have made revision in line 197-199, as that “By contrast, due to the backwardness of new energy technologies, countries with low R&D levels may face a huge economic cost of achieving new energy consumption transformation.”
Point 24: “…costs are mainly borne…” This phrase is not clear.
Response 24: Thanks for the suggestion. According to this point, we have made revision in line 204-223, as that “Some studies have indicated that the regional heterogeneity of the impact of new energy consumption transformation on economic growth is caused by the level of real GDP per capita. Therefore we assume the level of economic development is one of the reasons for the economic costs of new energy consumption transformation. Batlle (2011) [38] found that new energy development costs are high and the government needs to support preferential policies, such as subsidies, taxes, and loans. When the government revenue is certain, the expenditure on new energy subsidies will have a crowding out effect on other expenditures, which may be detrimental to the country's economic growth. If the new energy development expenditure is excessive, the powerful crowding out effect may cause enormous economic costs. However people in countries with high levels of economic development are generally advanced in environmental protection and have high demand for new energy consumption. Thereby people in countries with high economic development levels can spontaneously promote the transformation of new energy consumption, which can reduce the government's expenditure for new energy development. In addition, countries with high levels of economic development are prone to introducing the “combined effects”, thereby easily attracting capital, technology, and high-tech talent for new energy development, which can also help reduce the economic cost of new energy consumption transformation. Hence, this study proposes hypothesis 2: When the level of economic development is below a certain threshold, the impact of new energy consumption transformation on economic growth is negative. When the level of economic development is above a certain threshold, the impact may change from negative to positive.”
Point 25: “…to new energy sources…” instead of “…to ne energy sources…”
Response 25: Thanks for the suggestion. We apology for the mistake. We have made revision in line 239-240, as that “In these countries, producing economies of scale and learning effects is difficult in relation to new energy sources.”
Point 26: “…Unruh (2000) determined that the industrial economy has been locked in by traditional fossil fuel-based energy sources [27]”. This reference is not recent.
Response 26: Thanks for the suggestion. We have made revision in line 240-244, as that “Klitkou et al. (2015) [43] determined that mature energy technologies have a distinct advantage over new energy technologies, not because they are necessarily better, but because their positive feedback mechanisms create additional benefits and decrease costs of production. In other words, new energy technologies may increase the economic cost of production.”
Point 27: All the equations parameters should be explained within the text.
Response 27: Thanks for the suggestion. We have checked all the acronyms in this paper to ensure there is no unexplained equations parameters in the revised manuscript, please see page 8 and 9.
Point 28: “Dependent variable” and “explanatory variable”. These terms are not adequately explained.
Response 28: Thanks for the suggestion. We use “independent variable” to instead of “explanatory variable” and explain how to choose these variables. The revision in line 258-268, as that “Dependent variable: Dependent variable is used in statistics as the outcome variable which one can be predicted by using independent variables. Because the economic growth is the outcome variable of our study, so we use economic growth as dependent variable. Economic growth(Y, unit: millions of US dollars) is expressed in terms of actual Gross Domestic Product (GDP) of each country;
Independent variable: Independent variable is used in statistics to predict or explain dependent variable. We want to predict or explain economic growth through the level of new energy consumption transformation, therefore we use the level of new energy consumption transformation as independent variable. The level of new energy consumption transformation (new, unit: percentage) is expressed as the ratio of new energy consumption (unit: Million toe) to total energy consumption (unit: Million toe)”
Point 29: “All data (except for those in units of percentage) are logarithmically used to reduce the possible heteroscedasticity in the data”. This sentence should be further explained.
Response 29: Thanks for the suggestion. We have made revision in line 282-284, as that “All data are logarithmically used to reduce the possible heteroscedasticity in the data except for those in units of percentage (Because the data larger than 0 and less than 1 does not have extreme outliers, there is no need to eliminate heteroscedasticity).”
Point 30: Table 1. Units are missing.
Response 30: Thanks for the suggestion. We have redesigned Table 1, please see page 8.
Point 31: Eq. 1. All the equation symbols should be written in the same way. E.g. “xit” and “xit”.
.Response 31: Thanks for the suggestion. We have rewritten all the equation symbols in the same way. E.g. “xit”
Point 32: “…R&D (R&D)…” This is not correct.
Response 32: Thanks for the suggestion. We have made revision in line 272, as that “Level of R&D (unit: percentage)”
Point 33: Table 2 should not be placed at the beginning of section 5.
Response 33: Thanks for the suggestion. We placed Table 2 in the main text behind the first time they are cited, please see page 10.
Point 34: “We assume that the model has no thresholds, the model has a threshold, the model has two thresholds, and model (2) is estimated on the basis of the preceding three cases”. This sentence is not clear.
Response 34: Thanks for the suggestion. We have made revision in line 324-326, as that “In the context of equation (7) there are three cases: no thresholds (neithernorexist), the model has a threshold ( exists and does not exist), the model has two thresholds ( and are both exist), and model (7) is estimated on the basis of these three cases.”
Point 35: “(1) when the threshold variable lower than the low threshold value…” This sentence is not clear.
Response 35: Thanks for the suggestion. We have made revision in line 335-339, as that “The model with two threshold values is divided into the following three segments: (1) when the level of threshold variable is lower than the first threshold value, (2) when the level of threshold variable is higher than the first threshold value but lower than the second threshold value, and (3) when the level of threshold variable is high than the second threshold value.”
Point 36: “This study first proposed three hypotheses”. What do the authors mean?
Response 36: Thanks for the suggestion. We removed this sentence because of the substantial revision of the conclusion section.
Point 37: In this section, the original contribution of the research has to be presented by focusing on the research results based on the research questions.
Response 37: Thanks for the suggestion. We have made substantially revises to the conclusions for presenting the contribution of the article based on the research results.
In line 418-426, we have summarized an overall conclusion as that “This paper has researched the threshold effects of new energy consumption transformation on economic growth .Using panel data for seven countries with high levels of energy consumption (i.e., the US, China, Japan, Canada, South Korea, Germany, and France) over the period 1997-2016 to establish the panel threshold models. The results show that the threshold effects of the transformation of new energy consumption on economic growth is caused by the level of R&D, economic development and traditional energy dependence .The overall conclusion is that: the impact of new energy consumption transformation on economic growth is non-linear and it will change with the levels of R&D, economic development and traditional energy dependence are at different ranges of levels.”
In line 427-435, we have summarized a conclusion when the R&D level is used as a threshold variable, as that “when the R&D level is below than 0.009, the new energy consumption transformation will have a great negative impact on economic growth. When the level of R&D is higher than 0.009 but lower than 0.014, the negative impact of the new energy consumption transformation on economic growth will be weakened. When the level of R&D is higher than 0.014, the new energy consumption transformation will have a positive impact on economic growth. This conclusion indicates that the transformation of new energy consumption is not conducive to economic growth in countries with low R&D levels. On the contrary, the transformation of new energy consumption in countries with high R&D levels will promote economic growth.”
In line 436-444, we have summarized a conclusion when the level of economic development is used as a threshold variable, as that “When the level of economic development is below than 7.469, the new energy consumption transformation will have a negative impact on economic growth. When the level of economic development is higher than 7.469 but lower than 8.996, the new energy consumption transformation will have a positive impact on economic growth. When the level of economic development is higher than 7.469, the positive impact is strengthened. This conclusion indicates that in countries with low levels of economic development, the transformation of new energy consumption will have a negative impact on economic growth. However, in countries with high levels of economic development, the transformation of new energy consumption can promote economic growth.”
In line 445-452, we have summarized a conclusion when the level of traditional energy dependence is used as a threshold variable, as that “When the level of traditional energy dependence is below than 0.5695, the new energy consumption transformation will have a positive impact on economic growth. When the level of traditional energy dependence is higher than 0.5695, the new energy consumption transformation will have a negative impact on economic growth. This conclusion indicates that the transformation of new energy consumption will have a negative impact on the economic growth of countries with high dependence on traditional energy sources. However, the transformation of new energy consumption in countries with low levels of traditional energy dependence can promote economic growth.”
In order to couple the conclusions and recommendations more suitably, we have made revision in line 459-462, 471-475 and 483-486 as that
“First, countries with high R&D levels can accelerate the transformation of new energy consumption. However, countries with low R&D levels should give priority to improving R&D levels before new energy consumption transformation. Because high R&D levels can ensure that the country has sufficient technical support for new energy consumption transformation.”
“Second, countries with high levels of economic development can raise the level of new energy consumption, which is not only conducive to sustainable development, but also promotes national economic growth. Countries with low levels of economic development should be cautious in the transformation of new energy consumption, as this may be detrimental to the country’s economic growth.”
“Finally, for countries or regions with low levels of traditional energy dependence, the rate of new energy consumption transformation can be appropriately accelerated. For countries or regions where energy consumption has been “locked in” to fossil energy consumption, the transformation of new energy consumption should be approached with caution.”

Round 2
Reviewer 2 Report
In the revised edition, the manuscript has been significantly improved and covered the main comments of the reviewers.
However, I think that further editing is needed.
Some examples:
Abstract: [Lines 23-25]. This sentence needs to be improved.
“Hypothesis” or “hypothesis”?
References mentioned in the text should be improved
1. Introduction
[Lines 43-44]. This sentence needs to be improved.
Fig. 1 should be mentioned in the text.
[Lines 72-73]. This sentence needs to be improved.
2. Literature review
[Line 157]. “Some studies believed”?
Author Response
Response to Reviewer
Fangming Xie, Chuanzhe Liu, Huiying Chen and Ning Wang
At first, we appreciate editors’ and reviewers’ valuable comments and suggestions which help us improve the paper significantly. Following these comments and suggestions, we revised the paper as follows.
Response to Reviewer Comments
Point 1: Abstract: [Lines 23-25]. This sentence needs to be improved.
Response 1: Thanks for the suggestion. According to this point, we have revised the explanation of the threshold variable in line 23-26, as that “Threshold variable is essential in a panel threshold model. The behavioral varies of model can be predicted when threshold variable is at different ranges of levels. In other words, the behavior of panel threshold model may change as the level of threshold variable changes.”
Point 2: “Hypothesis” or “hypothesis”?
Response 2: Thanks for the suggestion. We apology for the mistake. We have used “Hypothesis” uniformly. We have revised in line 221,241 and 268, as that “Hypothesis”
Point 3: References mentioned in the text should be improved
Response 3: Thanks for the suggestion. We apology for the mistake. We have revised the format of all references by following the journal template. In addition, we have revised the references mentioned in the text in more detailed introduction, please see section 2.
Point 4: [Lines 43-44]. This sentence needs to be improved.
Response 4: Thanks for the suggestion. According to this point, we have revised the explanation of the new energy in line 44-47, as that “New energy is any energy source that is an alternative to fossil fuel. Therefore new energy is considered an environment-friendly energy source and intend to address concerns about fossil fuels, such as its high carbon dioxide emissions, an important factor in global warming.”
Point 5: Fig. 1 should be mentioned in the text.
Response 5: Thanks for the suggestion. We apology for the mistake. We have made revision in line 56, as that “As shown in Figure 1, the proportion of new energy consumption will increase rapidly,”
Point 6: [Lines 72-73]. This sentence needs to be improved.
Response 6: Thanks for the suggestion. According to this point, we have made revision in line 75-79, as that “Answering these questions can provide scientific recommendations for better development of new energy, avoiding the adverse effects of new energy consumption transformation on the economy and achieving the goal of new energy consumption share in Paris Agreement with minimal economic costs.”
Point 7: [Line 157]. “Some studies believed”?
Response 7: Thanks for the suggestion. This is the mistake of our expression. We apology for the mistake. We have made revision in line 178, as that ” Several studies indicated that”
